# Supplementation with Rumen-Protected Methionine Reduced the Parasitic Effect of *Haemonchus contortus* in Goats

**DOI:** 10.3390/vetsci10090559

**Published:** 2023-09-05

**Authors:** Laura Montout, Lahlou Bahloul, Dalila Feuillet, Max Jean-Bart, Harry Archimède, Jean-Christophe Bambou

**Affiliations:** 1Inrae, Asset, 97170 Petit Bourg, Guadeloupe, France; laura.montout@inrae.fr (L.M.); dalila.feuillet@inrae.fr (D.F.); harry.archimede@inrae.fr (H.A.); 2Centre of Expertise and Research in Nutrition, Adisseo France S.A.S., 2 Rue Marcel Lingot, 03600 Commentry, France; lahlou.bahloul@adisseo.com; 3Inrae, Plateforme Tropicale d’Expérimentation sur l’Animal, 97170 Petit Bourg, Guadeloupe, France; max.jean-bart@inrae.fr

**Keywords:** amino acids, nutrition, gastrointestinal parasite, goats

## Abstract

**Simple Summary:**

In this study, we investigated the impact of rumen-protected methionine supplementation on the response of Creole goat kids to an experimental infection with *Haemonchus contortus*, a gastrointestinal parasite that affects small ruminants. We found that while serum IgA anti-ESP was increased in methionine-supplemented animals, no significant effect on the level of parasitism was observed. However, our results suggested that methionine supplementation mitigated the detrimental pathophysiological impact of the infection. This study highlights the need for further research on ruminants to better understand how amino acids can influence the host’s immune response to parasitic infections. This knowledge could potentially lead to new strategies for controlling these infections in livestock, benefitting both animal health and welfare and production performances.

**Abstract:**

The present study investigated the impact of rumen-protected (RP) methionine supplementation on the resistance and resilience to *Haemonchus contortus* experimental infection of goat kids. Twenty-seven 6-month-old goat kids (14.55 ± 2.7 kg body weight) were placed in individual pens during an experimental period of forty-two days. Each kid was placed under one of three distinct diets (*n* = 9 animals/diet) corresponding to the following experimental groups: Control (C, Hay + concentrate), Low Methionine (LM, Hay + concentrate + 3.5 g/Kg of Dry Matter (DM) of RP methionine, or High Methionine (HM, Hay + concentrate + 11.5 g/Kg of DM of RP methionine). After a 4-week period of adaptation to the diets, all the animals were experimentally infected with a single oral dose of 10,000 *H. contortus* third-stage infective larvae (L3). No significant effect of RP methionine supplementation was observed for feed intake, digestibility and growth performance. The faecal egg counts (FEC) and worm burdens were not impacted by RP methionine supplementation either. In contrast, Packed cell volume (PCV) and haemoglobin concentration were higher in kids supplemented with RP methionine. Similarly, the level of serum IgA directed against adult *H. contortus* excretion and secretion products (ESP) was higher in supplemented kids. These results suggested that RP methionine supplementation improved goat kids’ resilience against *H. contortus* infection.

## 1. Introduction

Gastrointestinal nematode (GIN) infections are one of the major pathogenic constraints on efficient grazing ruminant production systems [1]. These infections are ubiquitous worldwide in regions where ruminants are raised. In the Caribbean, the prevalence of these nematodes varies among studies and species, with reported rates ranging up to 100% [2]. The worldwide emergence of anthelmintic-resistant strains and the associated animal welfare concerns have led to the development of complementary and more sustainable control strategies. Nutritional supplementation, particularly in metabolizable protein, to improve the host immune response and reduce the deleterious pathogenic impacts has long been considered one of these promising strategies [3]. Indeed, the host’s ability to develop effective immunity against pathogens, including GIN, is closely associated with nutritional status [4,5,6]. The response against invading pathogens is costly in energy and protein, notably due to the increase in the metabolic activity of immune cells for the synthesis of immune mediators, proliferation and repair of damaged tissue [7,8]. Minerals, trace elements and vitamins are also necessary for the development of immunity [9,10]. Despite the complexity of addressing experimentally the respective effects of energy and protein supplementation in ruminants, the current consensus is that the response against GIN is more sensitive to a deficiency in metabolizable protein than in metabolizable energy [11]. Moreover, the composition of microbial proteins from ruminal syntheses would not meet the needs of the immune system compared to that of dietary proteins (i.e., by-pass proteins) [11]. Indeed, recently, it has been shown in lambs and kids that diets enriched with rumen-protected protein (RPP) were associated with better resistance and resilience to GIN infection [12,13]. The difference between by-pass and microbial proteins, in terms of amino acid composition, would be implicated [11,14].

In a recent systematic review, it was shown that, in livestock animals, dietary supplementation with three synthetic amino acids in particular (i.e., methionine, threonine and arginine) was associated with a significant improvement in the immune response against infectious diseases [15]. However, most of the studies were conducted in poultry, so this review underlined the need to investigate this strategy in other livestock species. Thus, the objective of the present study was to investigate the effect of rumen-protected (RP) methionine supplementation on the response of kid goats against an experimental *Haemonchus contortus* infection.

## 2. Materials and Methods

### 2.1. Animals, Management and Experimental Design

This experiment was conducted at the INRAE PTEA (Plateforme Tropicale d’Expérimentation sur l’Animal) experimental farm in Guadeloupe (16°20′ North latitude, 61°30′ West longitude). All the animals owned by the INRAE PTEA have been reared at this experimental farm since 1980.

A total of 27 Creole goat male kids (14.55 ± 2.7 kg body weight (BW); 6 months old) were randomly chosen from the 12 sires’ families of the INRAE flock, which are raised on pasture in a rotational grazing system. The kids were drenched with levamisole (Polystrongle^®^, Coophavet, Ancenis, France, 8 mg/kg BW), toltrazuril (Baycox ovis^®^, Bayer healthcare, Lille, France, 20 mg/kg BW) and albendazole (Valbazen^®^ 1.9%, Zoetis, Paris, France, 7.5 mg/kg BW) and then were housed indoors under worm-free conditions. After a one-week period in groups, the kids were randomly placed in individual pens in which they could hear and see each other. During this period, nematode faecal egg counts (FEC) remained at zero. Each animal was placed under one of three distinct diets (*n* = 9 animals/diet) corresponding to the following experimental groups: Control (C, Hay + concentrate), Low Methionine (LM, Hay + concentrate + 3.5 g/Kg of Dry Matter (DM) of RP methionine (smartamine^®^, Adisseo, Anthony, France), or High Methionine (HM, Hay + concentrate +11.5 g/Kg of DM of RP methionine), and had free access to fresh water (Table 1). The RP methionine was distributed to the animals mixed with pellets of the concentrate before the hay. The ingredient and chemical composition of the experimental diets is shown in Table 1. After 4 weeks of adaptation to the diets, all the animals were experimentally infected with a single oral dose of 10,000 *H. contortus* third-stage infective larvae (L3). The experiment lasted a total of 70 days: 28 days before and 42 days after the experimental infection.

The L3 were obtained 54 days before the experimental infection from coproculture of monospecifically infected donor Creole goats with isolates previously obtained from Creole goats reared on pasture in different farms in Guadeloupe.

### 2.2. Animal Samples and Measurements

The animals were weighed weekly to individually adjust the offered quantities of feed at 3% of the BW and to measure individual growth rates. From the day of experimental infection (day 0 post infection, dpi), blood samples were collected at a 7-day interval by jugular venipuncture on each animal using disposable syringes and 20-Ga needles in tubes containing an anticoagulant for complete blood counts (BD Vacutainer^®^ spray-coated K3EDTA, Becton, Dickinson and Company, Franklin Lakes, NJ, USA) and in dry tubes for serum analysis (BD Vacutainer^®^, Becton, Dickinson and Company, Franklin Lakes, NJ, USA). Blood samples were analyzed by an automaton (Melet Schloesing, MS9-5s, Osny, France). The number of circulating eosinophils was determined using a Malassez cell counter. For the serological analysis, blood samples from each animal were centrifuged for 5 min at 5000 rpm, then 2 aliquots of 1 mL of serum/animal were frozen at −20 °C until analysis. Serum pepsinogen levels were determined according to the method of Dorny and Vercruysse [16]. The FEC measurements were performed on approximately 10 g of faeces collected in plastic tubes directly from the rectum of each animal and transported to the laboratory in refrigerated vials. The samples were analyzed individually using a modified McMaster method for rapid determination, and FEC was expressed as the number of eggs/g faeces [17]. The faeces were collected twice daily from Monday to Friday in collection bags that had been fitted to each animal and weighed. At slaughter (42 d.p.i), the contents of the abomasum of the infected animals were collected individually. Thereafter, to recover all the established parasites, the abomasum was washed with warm distilled water and scraped with a microscope slide. The contents and the wash water were stored at 4 °C until counting. The parasites were collected, counted and sorted according to the method as previously described [18]. The levels of IgA and IgE anti-L3 (crude extract of larval stage 3 antigens) and anti-ESP (Excretion Secretion Products of adult *H. contortus*) were measured by indirect ELISA according to Bambou et al. [17]. Crude extracts and ESP of *H. contortus* L3 and adults, respectively, were prepared according to Bambou et al. [17]. In order to compare results between assays, a positive control consisting of a pool of sera containing IgA and IgE antibodies was included on each plate, and the OD450 measurements of unknown samples were altered in proportion with changes in this standard. To measure the animal’s ingestion and digestibility during 2 periods of 5 consecutive days (between 21 and 26 dpi, then 35 and 40 dpi), all feed offered and refused was individually weighed and sampled. During these periods, the whole faeces excretions were individually measured and sampled. Daily, samples were proportionally pooled and preserved.

### 2.3. Chemical Analyses and Analytical Procedures

The DM contents of feed and faeces were determined by oven drying (Type SE-79, Le Matériel Physico Chimique Flam et Cie, Neuilly-sur-Marne, France) to a constant weight at 60 °C for 48 h, while ash content was determined by heating samples at 550 °C for 4 h according to AOAC [19,20]. The organic matter (OM) content was calculated by difference. Dry samples were obtained for further chemical analyses and were ground (model SK100 confort Gußeisen, F. Kurt Retsch GmbH & Co, Haan, Germany) to pass through a stainless-steel screen (1 mm). The crude protein (CP) content was calculated after N determination by combustion using the micro-Dumas method (CE Instruments, ThermoQuest S.p.A., Milano, Italy).

### 2.4. Statistical Analysis

The parameters measured were analyzed by a linear mixed model using the Proc Mixed of the software SAS (version 9.4 TS Level 1M3). Because of the skewed distributions, FEC and eosinophil variables were logarithm transformed in Log (FEC + 15) and log (eosinophils + 1), respectively, to normalize the data. The other haematological data were square-root transformed to normalize residual variances. The model included fixed effects of days post infection, group and significant interaction. For all the analysis, the Shapiro–Wilk test indicated that the residuals were normally distributed. The comparisons between means were conducted by the least square means procedure. Means were considered significantly different for *p* ≤ 0.05.

## 3. Results

### 3.1. Experimental Diets and Growth Performance

The control, Low Methionine and High Methionine experimental diets contained, respectively (/kg dry matter): 921.9 g, 921.6 g and 920.7 g of organic matter, 75.2 g, 74.9 g and 75.4 g of crude protein and 9.25 MJ/Kg of metabolizable energy (Table 1). The means of the ADG of the three experimental groups range between 27.3 and 36.7 g/d, but no difference between groups was observed (Table 2). No difference was observed for the intake between groups, except for the CPI (crude protein intake), which was higher during period 2 compared with period 1, whatever the group (39.8 and 34.3 g/d respectively, *p* = 0.01, Table 2). The digestibility of the dry matter (DDM) and the organic matter (DOM) were significantly higher during period 1 compared with period 2 (676.5 and 651.5 g/Kg of DM and 691.4 and 671.4 g/Kg of DM, respectively, *p* < 0.05, Table 2). For the digestibility of the crude protein, the mean was higher for period 1 compared with period 2 (463.7 and 584.5 g/Kg of DM, respectively). The means of the ADG of the three experimental groups range between 27.3 and 36.7 g/d, representing + 34.4%, but no difference between groups was observed (Table 2).

### 3.2. Parasitological and Physiological Parameters

No significant difference was observed between the experimental groups for the means of FEC, female and male adult worms and parasite burden (*p* > 0.05, Table 3). The packed cell volume (PCV) and the haemoglobin blood concentration decreased significantly, whatever the group, from 0 to 21 dpi (*p* < 0.01, Figure 1 and Figure 2). No effect of the group was observed, but there was a significant interaction between dpi and groups showing significant difference from 21 to 42 dpi. The PCV and the haemoglobin blood concentration were significantly lower in the C group compared to the LM and the HM ones. The PCV and the haemoglobin blood concentration of the HM group decreased slightly between 14 and 21 dpi, then at 35 and 42 dpi, the kids recovered physiological values (0 dpi). In contrast, the values of the kids from the LM groups remained significantly lower, at 35 and 42 dpi, than physiological values. The platelet concentrations decreased from 0 to 42 dpi, whatever the group, and were higher in the supplemented groups (LM and HM) compared to the C group, whatever the timepoint (*p* < 0.0001, Figure 3).

### 3.3. Serological and Blood Analysis

A significant effect of the dpi was observed for all the immunoserological parameters analyzed (*p* < 0.05, Table 4). However, no interaction between the groups and the dpi was observed, whatever the serological parameters (*p* > 0.05, Table 4). The level of IgE anti-L3 was significantly higher in the C group compared with the LM (*p* = 0.0043), but no difference was observed with the HM and between the HM and the LM. No effect of the groups was observed for the level of IgE anti-ESP. For the IgA anti-L3 and anti-ESP, the levels of the HM group were significantly higher than the C groups. The level of IgA anti-ESP was higher in the LM compared with the C group (*p* = 0.0029), but no difference was observed for the level of IgA anti-L3. The level of serum pepsinogen was higher in the C and LM groups compared with the HM one (*p* = 0.005).

The same results were observed for the blood cell counts, with a significant effect of the dpi for all the parameters analyzed (*p* < 0.01, Table 5), except for the percentage of blood eosinophils, and no interaction between the groups and the dpi (*p* > 0.05, Table 4). No significant effect of the groups was observed for the percentage of blood lymphocytes and monocytes, but the percentages of blood neutrophils and eosinophils were higher in the HM group than in the C and LM ones (*p* < 0.004, Table 5).

## 4. Discussion

The management of small ruminant nutrition has long been considered a tool for controlling GIN infections [21,22]. Protein supplementation, especially with by-pass protein, could potentially improve the host response to GIN by fostering the development of an efficient protective immune response and/or reducing the pathophysiological consequences of the infection, depending on their specific amino acid composition [11]. Indeed, the profile and the quantity of amino acids absorbed in the small intestine can be influenced by the composition of feed protein undegraded in the rumen. By-pass protein’s amino acid composition may be more compatible with that of immune proteins. However, while supplementation with rumen-protected protein and amino acids to support maintenance and productive functions in ruminants is extensively studied, its potential as a tool for the non-chemical control of GIN infection has been poorly investigated [23,24].

In a recent systematic review, we showed that sulphur-containing amino acids, such as methionine, improved the host immune response against infectious diseases in livestock [15]. This review also revealed that the majority of studies were conducted in poultry, underscoring the necessity for more research in ruminants. Therefore, the objective of the current study was to assess the impact of rumen-protected methionine supplementation on the response of Creole goat kids to an experimental *H. contortus* infection.

Numerous studies have investigated the effect of supplementation with rumen-protected methionine, mainly on milk production and to a lesser extent on growth performance, and showed an increase in feed intake, milk yield and milk protein and growth rate [23,25]. Most of these studies compared control diets with diets supplemented with methionine at different levels (control diet + rumen-protected methionine). In the present study, the diets were iso-energetic and iso-nitrogenous in order to investigate the specific effect of a higher level of methionine and not the cumulative effect with a higher level of nitrogen. Thus, here we showed that rumen-protected methionine supplementation has no impact on feed intake, digestibility and growth rate of kid goats infected with *H. contortus*. In line with this result, it has been reported that rumen-protected methionine supplementation did not improve dry matter digestibility and nitrogen utilization of growing beef steers challenged with a Gram-negative bacterial endotoxin [26]. However, non-infected lambs supplemented with rumen-protected methionine showed increased dry matter intake and growth rate [27,28].

Recently, two studies in kid goats and lambs, challenged either to experimental infection with *H. contortus* or to natural multispecies infection at pasture, showed that diets supplemented with RPP or rumen undegradable protein (RUP) in goats and lambs, respectively, significantly reduced the level and the detrimental impact of parasitism as measured through FEC and PCV [12,13]. Our results showed that FEC and nematode burden were not affected by rumen-protected methionine, regardless of the level of supplementation. Nonetheless, it appears that the specific AA composition of by-pass proteins could contribute to reducing the amount of eggs excreted. In contrast, rumen-protected methionine supplementation demonstrated a mitigating effect on the detrimental consequences of *H. contortus* infection. Indeed, after the experimental infection, the PCV and the haemoglobin concentration decreased, indicating anaemia caused by haemonchosis, but it remained higher in the supplemented groups. Furthermore, serum pepsinogen concentrations, an indicator of the severity of mucosal lesions caused by GIN, remained lower in the supplemented groups. In accordance with these results, the thrombocytopenia typically observed during *H. contortus* infection was less severe in the supplemented groups. Altogether, these results strongly suggest that rumen-protected methionine supplementation induced resilience to *H. contortus* infection in growing kid goats. As hypothesized earlier, the resilience induced by dietary proteins could be attributed to the compensation for the loss of nutrients that occurs during GIN infection [29].

Protein supplementation has also been linked to host resistance to GIN, resulting in decreased FEC and worm burdens, especially during the phase of expression of immunity [24]. In this study, the percentages of peripheral eosinophils and neutrophils in response to GIN infection were lower in the HM group compared to the C and LM groups. Recently, polymorphonuclear neutrophils and eosinophils have been described as essential host effectors involved in the mechanisms that limit the establishment of *H. contortus* [30]. In contrast to sheep, several studies conducted in goats have not found a direct relationship between blood eosinophilia and the level of parasitism [31,32,33]. Indeed, it has been suggested that the eosinophil response to *Teladorsagia circumcincta* would be less effective in goats compared to sheep [34]. The authors of this study also performed molecular modelling and identified a dysfunctional transmembrane domain of the high-affinity IgA receptor, suggesting that the IgA response might not be effective in goats. However, our findings demonstrate an increase in serum IgA anti-ESP in kid goats supplemented with rumen-protected methionine. Accordingly, the supplementation led to an enhanced IgA response against *H. contortus*, resulting in reduced female worm prolificity in kid goats [12]. In this study, no significant correlation was found between the IgA response and the prolificity of *H. contortus*. In this Creole breed, previous studies have indicated that the IgE response against *H. contortus* L3 may be one important pathway contributing to the development of host resistance to GIN infections (i.e., reduction of the level of parasitism) [35]. In the present study, no effect on the level of parasitism was induced by the diet supplemented with rumen-protected methionine. Consequently, no impact of the supplementation was observed on the IgE response.

In conclusion, our study demonstrated that rumen-protected methionine supplementation did not influence the level of parasitism in kid goats. Despite an increase in serum IgA anti-ESP, no significant effect on the IgE response was observed. These findings contribute interesting insights to the understanding of host–parasite interactions in goats, contributing to future research aimed at optimizing strategies to enhance host resistance/resilience against GIN and improve livestock health.

## Figures and Tables

**Figure 1 vetsci-10-00559-f001:**
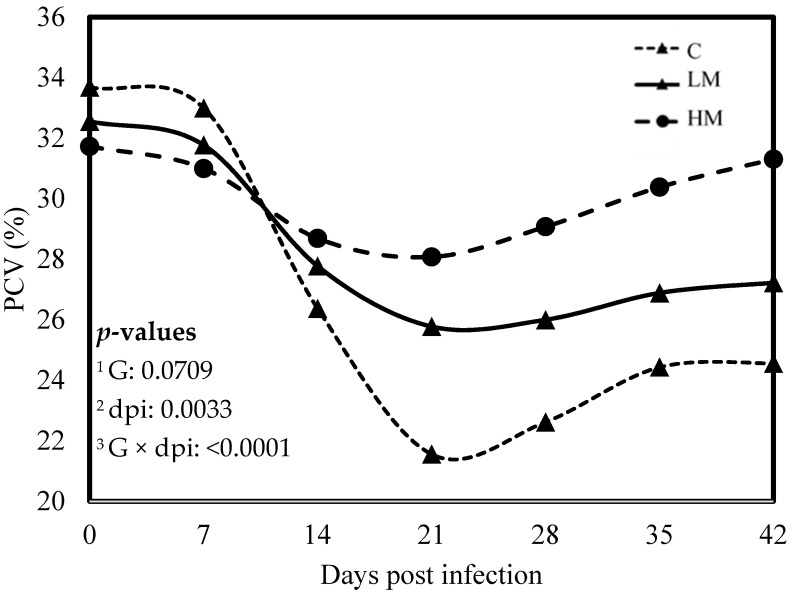
Least square means of packed cell volume (PCV) of Creole kids according to the three experimental groups (C, Control; LM, Low Methionine; and HM, High Methionine) infected with an oral single dose of 10,000 third-larvae stage (L3) of *Haemonchus contortus*. ^1^ G: experimental groups, ^2^ dpi: days post infection, ^3^ G × dpi: interaction between G and dpi.

**Figure 2 vetsci-10-00559-f002:**
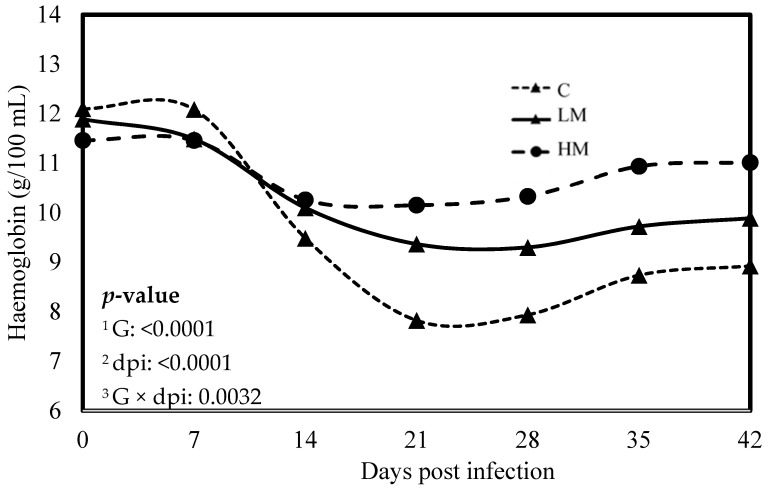
Least square means of hemoglobin concentration of Creole kids according to the three experimental groups (C, Control; LM, Low Methionine; and HM, High Methionine) infected with an oral single dose of 10,000 third-larvae stage (L3) of *Haemonchus contortus*. ^1^ G: Experimental groups of treatment, ^2^ dpi: days post infection, ^3^ G × dpi: interaction between G and dpi.

**Figure 3 vetsci-10-00559-f003:**
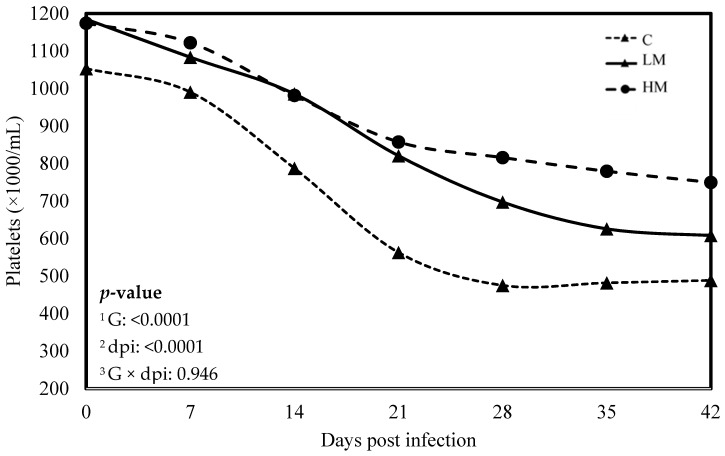
Least square means of platelet concentration of Creole kids according to the three experimental groups (C, Control; LM, Low Methionine; and HM, High Methionine) infected with an oral single dose of 10,000 third-larvae stage (L3) of *Haemonchus contortus*. ^1^ G: Experimental groups, ^2^ dpi: days post infection, ^3^ G × dpi: interaction between G and dpi.

**Table 1 vetsci-10-00559-t001:** Ingredient and nutrient composition of experimental diets.

Diet Composition	Control	Low Methionine	High Methionine
Ingredient composition (g/kg DM)			
Maize	221.5	225	230
Soybean Meal	103	96	83
Hay (*Digitaria* spp.)	675	675	675
Methionine (Smartamine^®^)	0	3.5	11.5
Mineral Mixture	0.5	0.5	0.5
Nutrient composition (g/Kg)			
Dry Matter	931.4	931.6	920.7
Organic Matter	921.9	921.6	920.7
Crude Protein	75.2	74.9	75.4
Neutral Detergent Fiber	537.2	536.0	533.7
Acid Detergent Fiber	304.9	304.3	303.1
Acid Detergent Lignin	53.9	53.9	53.8
Methionine	1.02	3.31	8.56
Predictive nutritive value			
Metabolizable Energy (MJ/Kg)	9.25	9.25	9.25
Net Energy (MJ/Kg)	5.47	5.47	5.48
PDI ^1^ (g/Kg)	85.66	85.65	85.67
PDIA ^2^	44.36	44.38	44.42
MetDi ^3^ (% PDI)	1.87	2.02	2.75
LysDi ^4^ (% PDI)	6.60	6.56	6.62
HisDi ^5^ (% PDI)	2.12	2.12	2.12

^1^ PDI: protein digested in the small intestine. ^2^ PDIA: dietary protein undegraded in the rumen and truly digested in the small intestine. ^3^ MetDi: digestible Methionine. ^4^ LysDi: digestible Lysine. ^5^ HisDi: digestible Histidine.

**Table 2 vetsci-10-00559-t002:** Least square means of intake and total tract digestibility of dry matter (DMI, DDM), organic matter (OMI, DOM), crude protein (CPI, DCP), neutral detergent fiber (NDFI) and acid detergent fiber (ADFI) of the three experimental groups during two periods of 5 days, 21 to 26 (period 1) and 35 to 40 (period 2) days post infection with 10,000 *H. contortus* L3.

	Period 1 21 to 26 dpi ^1^			Period 2 35 to 40 dpi				*p*-Value		
	C ^2^	LM ^3^	HM ^4^	C	LM	HM	SEM	G ^5^	P ^6^	G × P
Intake (g/d)										
DMI ^7^	490.5	543.2	506.0	475.8	516.1	527.0	12.46	0.102	0.697	0.505
OMI ^8^	438.3	489.0	457.8	430.2	471.1	477.5	11.20	0.064	0.895	0.599
CPI ^9^	32.9 ^a^	35.3 ^a^	34.1 ^a^	36.4 ^b^	38.3 ^b^	44.5 ^b^	1.48	0.189	0.010	0.270
NDFI ^10^	238.4	269.6	228.4	243.4	268.6	266.0	9.08	0.186	0.281	0.415
ADFI ^11^	137.1	152.9	135.3	139.9	152.7	150.7	5.19	0.286	0.418	0.649
Total tract digestibility (g/kg DM)										
DDM ^12^	671.4 ^a^	697.1 ^a^	660.9 ^a^	642.3 ^b^	656.1 ^b^	656.0 ^b^	6.92	0.199	0.014	0.313
DOM ^13^	684.9 ^a^	711.6 ^a^	677.6 ^a^	661.5 ^b^	679.3 ^b^	673.4 ^b^	6.44	0.105	0.033	0.431
DCP ^14^	428.0 ^a^	477.9 ^a^	486.8 ^a^	545.5 ^b^	572.6 ^b^	638.7 ^b^	22.92	0.166	0.001	0.765
ADG ^15^ (g/d)	27.3	35.2	36.7	27.3	35.2	36.7	4.38	0.417	1.000	1.000

Means with different superscripts within lines are significantly different. ^1^ dpi: Days post infection. ^2^ C: Control group fed with Hay + concentrate. ^3^ LM: Low Methionine group fed with Hay + concentrate + 3.5 g/Kg of Dry Matter (DM) of rumen-protected methionine. ^4^ HM: High Methionine group fed with Hay + concentrate + 11.5 g/Kg of Dry Matter (DM) of rumen-protected methionine. ^5^ G: Experimental groups. ^6^ P: Period, intake and total tract digestibility were measured during two distinct periods post infection (21 to 26 and 35 to 40 dpi). ^7^ DMI: Dry Matter Intake. ^8^ OMI: Organic Matter Intake. ^9^ CPI: Crude Protein Intake. ^10^ NDFI: Neutral Detergent Fiber Intake. ^11^ ADFI: Acid Detergent Fiber Intake. ^12^ DDM: Dry Matter Digestibility. ^13^ DOM: Organic Matter Digestibility. ^14^ DCP: Crude Protein Digestibility. ^15^ ADG: Average Daily Gain. Means with different superscripts within a line differ significantly (*p* < 0.05).

**Table 3 vetsci-10-00559-t003:** Least square means of faecal egg counts (FEC) from 21 to 42 days post infection (dpi) and adult parasite counts of the three experimental groups at 42 dpi with 10,000 *H. contortus* L3.

	C ^1^	LM ^2^	HM ^3^	SEM ^4^	*p*-Value		
					G ^5^	dpi ^6^	G × dpi
FEC (eggs/g of faeces)	5369	2773	3001	1652	0.366	<0.0001	0.577
Female	389	291	290	94	0.727	-	-
Male	349	297	202	161	0.653	-	-
Parasite burden	739	589	492	292	0.702	-	-

^1^ C: Control group fed with Hay + concentrate. ^2^ LM: Low Methionine group fed with Hay + concentrate + 3.5 g/Kg of Dry Matter (DM) of rumen-protected methionine. ^3^ HM: High Methionine group fed with Hay + concentrate + 11.5 g/Kg of Dry Matter (DM) of rumen-protected methionine. ^4^ SEM: Standard Error of the Mean. ^5^ G: Experimental groups. ^6^ dpi: days post infection.

**Table 4 vetsci-10-00559-t004:** Least square means of the humoral IgE and IgA response of the three experimental groups against *H. contortus* after experimental infection with 10,000 L3.

	C ^1^	LM ^2^	HM ^3^	SEM ^4^	*p*-Value		
					G ^5^	dpi ^6^	G × dpi
IgE anti-L3 ^7^	0.175 ^a^	0.141 ^b^	0.157 ^ab^	0.009	0.017	0.009	0.789
IgE anti-ESP ^8^	0.143	0.131	0.108	0.005	0.130	<0.0001	0.667
IgA anti-L3	0.183 ^a^	0.200 ^a^	0.225 ^b^	0.021	0.021	0.009	0.994
IgA anti-ESP	0.159 ^a^	0.255 ^b^	0.243 ^b^	0.024	0.008	0.013	0.984
U_Tyrosine ^9^	1.624 ^a^	1.521 ^a^	1.346 ^b^	0.061	0.005	<0.0001	0.952

Means with different superscripts within lines are significantly different. ^1^ C: Control group fed with Hay + concentrate. ^2^ LM: Low Methionine group fed with Hay + concentrate + 3.5 g/Kg of Dry Matter (DM) of rumen-protected methionine. ^3^ HM: High Methionine group fed with Hay + concentrate + 11.5 g/Kg of Dry Matter (DM) of rumen-protected methionine. ^4^ SEM: Standard Error of the Mean. ^5^ G: Experimental groups. ^6^ dpi: days post infection. ^7^ L3: Crude *H. contortus* third-stage (L3) larval extract. ^8^ ESP: Excretory secretory products of adult *H. contortus*. ^9^ U_Tyrosine: Serum pepsinogen levels in units of tyrosine (U Tyr/L of serum). Means with different superscripts within a line differ significantly (*p* < 0.05).

**Table 5 vetsci-10-00559-t005:** Least square means of circulating blood cells of the three experimental groups after experimental infection with 10,000 *H. contortus* L3.

Complete Blood Count (%)	C ^1^	LM ^2^	HM ^3^	SEM ^4^	*p*-Value		
					G ^5^	dpi ^6^	G × dpi
Lymphocytes	40.04	40.37	39.63	1.3	0.828	0.003	0.828
Monocytes	4.06	4.26	4.16	1.0	0.324	0.01	0.668
Polymorphonuclear neutrophils	47.15 ^a^	45.08 ^a^	51.95 ^b^	1.6	0.004	0.012	0.674
Eosinophils	8.61 ^a^	10.16 ^a^	4.17 ^b^	1.2	0.002	0.343	0.443

Means with different superscripts within lines are significantly different^. 1^ C: Control group fed with Hay + concentrate. ^2^ LM: Low Methionine group fed with Hay + concentrate + 3.5 g/Kg of Dry Matter (DM) of rumen-protected methionine. ^3^ HM: High Methionine group fed with Hay + concentrate + 11.5 g/Kg of Dry Matter (DM) of rumen-protected methionine. ^4^ SEM: Standard Error of the Mean. ^5^ G: Experimental groups. ^6^ dpi: days post infection. Means with different superscripts within a line differ significantly (*p* < 0.05).

## Data Availability

Detailed data are available from Jean-Christophe Bambou via the e-mail address shown on the title page.

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
