# Peer review of "Supplementation with Rumen-Protected Methionine Reduced the Parasitic Effect of Haemonchus contortus in Goats"

_vetsci, 2023, doi:10.3390/vetsci10090559_

Round 1

Reviewer 1 Report

In my opinion this manuscript will be and advance for the researchers and could be interest apply in small ruminants vets as well. 

Author Response

We thank the reviewer for his comment

Reviewer 2 Report

Dear Editor,

The study entitled as “Supplementation with rumen protected methionine reduced the parasitic effect of Haemonchus contortus in goats” explore the effect of feed added-amino acid in reduction of parasitic load. This is a good study need to be published after minor revision.

-      Need to improve English language.

-      How the authors have selected 27 kids?

-      What about the normality of you data before statistical analysis?

-      Please also mention some prevalence study in France, especially your study area.

-      In discussion part, the comparison of your study results with other published paper are nor much adequate, please fortify them.  

-      Please provide some future insight to your reader in last paragraph of discussion.

-      A separate conclusion is appreciated or otherwise the last paragraph of discussion should summarized the key results in conclusion.

-      Suggestion: need to cite some more relevant studies.

-      Where are the figures?

-      Need to improve English language.

Author Response

Dear reviewer,

We thank you for taking the time to review our manuscript. Please find below our answers to your comments and questions.

Best regards,

jcb

  1. Need to improve English language.

R/ The revised version of the manuscript has been thoroughly proofread by a colleague proficient in English.

  1. How the authors have selected 27 kids?

R/ Lines 72-73 of the revised manuscript: “A total of 27 Creole goat male kids (14.55 ± 2.7 kg body weight (BW); 6 months old) were randomly chosen from the 12 sires families of the INRAE flock, raised on pasture in a rotational grazing system”

  1. What about the normality of you data before statistical analysis?

R/ Corrected. Lines 129-130 of the revised manuscript: “For all the analysis, the Shapiro-Wilk test indicated that the residuals were normally distributed”

  1. Please also mention some prevalence study in France, especially your study area.

R/ Corrected. Lines 41-43 of the revised manuscript: “These infections are ubiquitous worldwide in regions where ruminants are raised (O’Connor et al. 2006). In the Caribbean, the prevalence of these nematodes varies among studies and species, with reported rates ranging up to 100% (Cameroon-Blake et al. 2022).

  1. In discussion part, the comparison of your study results with other published paper are nor much adequate, please fortify them.

R/ The discussion section has been revised to address this comment.

  1. Please provide some future insight to your reader in last paragraph of discussion.

R/Done

  1. A separate conclusion is appreciated or otherwise the last paragraph of discussion should summarized the key results in conclusion.

R/Done

  1. Suggestion: need to cite some more relevant studies.

R/ As mentioned in the manuscript, research focusing on the influence of amino acids on the response of ruminants to gastrointestinal nematodes (GIN) remains limited, and in the case of goats, it is even more sparse. However, we have thoroughly revised the discussion section to address this gap.

  1. Where are the figures?

R/Corrected

Reviewer 3 Report

The study has a certain degree of novelty and may open the way to new research in the field. The paper is hard to follow because there are a lot of abbreviation.

The discussion chapter must be corelated with the result obtained by the authors.

It is necessary to add a separate chapter of conclusions. 

Author Response

Dear reviewer,

We thank you for taking the time to review our manuscript. Please find below our answers to your comments and questions.

Best regards,

  1. The study has a certain degree of novelty and may open the way to new research in the field. The paper is hard to follow because there are a lot of abbreviation.

R/ We have addressed this concern by ensuring that all abbreviations used are clearly defined upon their first mention in the text and by striving for clarity and conciseness

  1. The discussion chapter must be corelated with the result obtained by the authors.

R/ The discussion section has been revised to address this comment.

  1. It is necessary to add a separate chapter of conclusions. 

R/Done